# Cognitive-Awakening Chain-of-Surgery for Compositional Zero-Shot Surgical Triplet Recognition

## Abstract

Compositional Zero-shot Surgical Triplet Recognition (CZSTR) is a challenging task that requires models to recognize unseen combinations of <*instrument, verb, target*> that never co-occurred during training. This task captures the inherent generalization requirement in real surgical procedures. Large Vision-Language Models (LVLMs) with Chain-of-Thought (CoT), as one of the most advanced methods, are limitedly exposed to sufficient surgical semantics, leading to a shortage on the CZSTR task. To tackle this, we explore a more intuitive and natural human-like reasoning framework, which is introduced as **Co**gnitive-awakening **Ch**ain-**o**f-**S**urgery (CoCoS). CoCoS mirrors the way surgeons think: it starts by glancing at the scene, then gazing at the operation process over time, and finally drawing structured conclusions. Such a step-by-step cognitive-awakening process reflects how we naturally interpret surgical procedures and instruct large vision-language models (LVLMs) to deeply understand surgical scenes. Observing that LVLMs often hallucinate on relatively simple subtasks, e.g., identifying instruments, we further propose a Multimodal image–Sequence–Text (MiST) fusion module to reinforce the stability of the framework. To evaluate our framework, we also develop a strategy to reorganize existing surgical triplet datasets into a compositional zero-shot benchmark. Experiments show that our framework improves generalization to unseen triplets, outperforming both traditional models and LVLMs under this challenging task.

## 1 Introduction

Surgical video understanding, as a critical aspect in the development of intelligent assistance systems for laparoscopic surgery, has seen a surge of interest in recent years from researchers(Hu et al., 2024; 2025a). In contrast to the coarse-grained phase recognition, recognizing fine-grained surgical actions in the form of structured <*instrument, verb, target*> triplets (Nwoye et al., 2023a;b) is essential for building interpretable, generalizable, and trustworthy models of surgical activity. Such fine-grained tasks are crucial for evaluating surgical quality and documenting procedure details.

Conventional visual recognition frameworks (Nwoye et al., 2022; Sharma et al., 2023) have demonstrated reliable performance in early benchmarks, but they often struggle with generalization, semantic abstraction, and compositional reasoning, especially in zero-shot scenarios. In recent years, methods based on Visual Language Model (VLM) (Li et al., 2025a; Chen et al., 2025; Xi et al., 2023; Sharma et al., 2025) have achieved impressive results, especially in aligning textual and visual representations, but most of them put more emphasis on text-side modeling, leaving the visual side underexplored. Methods such as (Low et al., 2025), which enhance implicit information in text generated by VLM, often underestimate temporal relations in continuous sequences with consistent labels.

To address the shortcomings mentioned above, we propose a novel framework called **Co**gnitive-Awakening **Ch**ain-**o**f-**S**urgery (CoCoS), as shown in Figure 1. Inspired by how surgeons observe and reason during and after operations, **Ch**ain-**o**f-**S**urgery (CoS) is designed to comprehend surgical video clips in a progressive, step-by-step manner.

Unlike previous methods using VLM to retrieve captions of every single given frame in a single pass, our Chain-of-Surgery presents frames and corresponding video clips separately during the glancing and gazing stages within a shared context, encouraging LVLMs to engage more deeply with the complex surgical scenes. To aggregate multimodal information more effectively, we introduce the Multimodal image–Sequence–Text (MiST) fusion module. MiST aggregates pre-observation during glancing, procedure evaluation during gazing, triplet judgement during thinking, image features, and video features, enabling comprehensive diagnosis.

In addition to architectural innovation, we address a zero-shot compositional surgical triplet recognition task to capture the inherent generalization requirement in real surgical procedures. Unlike prior work where all test triplets appear in training, we construct splits where each individual component *<instrument, verb, target>* is seen during training, but their specific combinations are unseen at test time. This setting closely mimics real-world deployment, where a surgical assistance system must reason about plausible yet unseen action compositions in the operating room.

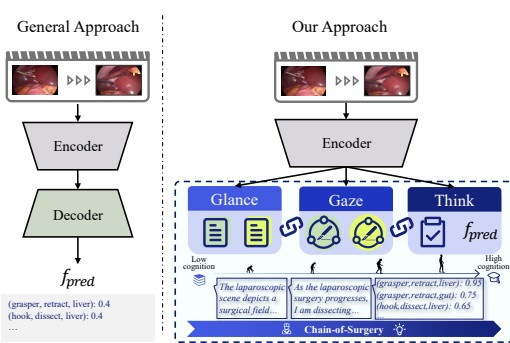

Figure 1: Insights of this work. General approach directly gives the score of surgical triplet recognition, suffering from the generalization of the compositional zero-shot case. Our approach understands surgical scenes via a human-like cognitive-awakening process: Glance, Gaze, and Think, showing better generalization of the compositional zero-shot case.

In summary, the contributions of this work are as follows:

- **Human-like reasoning framework.** We propose a novel **Co**gnitive-Awakening **C**hain-**of**-**S**urgery (CoCoS) framework to instruct LVLMs to perform surgical scene reasoning from low-level cognition to high-level cognition.

- **Chain-of-Surgery.** We present a chain-of-surgery prompting scheme that gradually shows visual data to the model, which constructs an accumulated context step by step. Chain-of-Surgery (CoS) helps LVLMs to focus more on surgical scenes and retrieve relevant domain knowledge.

- **Multimodal image–Sequence–Text fusion.** We develop MiST, a transformer-based fusion module that fuses visual features from images and videos with LVLM-encoded semantic descriptions. By leveraging information across spatial, temporal, and textual domains, MiST serves as a bridge for triplet diagnosis.

## 2 RELATED WORKS

### 2.1 SURGICAL ACTION TRIPLET RECOGNITION

Understanding surgical activities at a fine-grained level is crucial for surgery AI assistance systems. Recent works have introduced triplet-based modeling, which simultaneously recognizes triplets *<instrument, verb, target>* in endoscopic videos. RDV (Nwoye et al., 2022) pioneered this direction by defining the recognition of surgical actions as triplet classification tasks, using a structured learning framework to predict action combinations such as *<grasper, grasp, liver>*. Some studies (Lin et al., 2024; Pei et al., 2025; Liu et al., 2024) try to capture the interactions between surgical instruments and anatomical targets explicitly which leverage spatial-temporal attention to condition tool usage on nearby anatomy and temporal context. From the perspective of long-tailed data distribution, some multi-task learning and contrastive methods (Li et al., 2023; Gui & Wang, 2024) have been proposed to improve the model performance on rare triplet categories. Despite these innovations, existing surgical triplet recognition models tend to struggle when generalizing to unseen combinations due to their closed-world assumptions. This limitation motivates the transition to compositional generalization, which we address through a zero-shot lens.

## 2.2 COMPOSITIONAL ZERO-SHOT RECOGNITION

Compositional Zero-Shot Learning (CZSL) aims to recognize novel attribute–object combinations unseen during training, making it suitable for surgical settings where plausible but unobserved tool–action–target triplets often arise. The foundational idea of the visual composition concept (Misra et al., 2017) was introduced to enable models to infer valid novel pairs by learning from seen ones. Building on this, (Naeem et al., 2021) proposed a graph embedding framework that leverages structured semantic graphs to capture higher-order dependencies and guide composition in zero-shot settings. More recently, with the extensive exploration of CLIP, many researchers have adapted CLIP to the CZS tasks through graph modeling, soft prompting, and particular prompt structure designing (Nayak et al., 2022; Li et al., 2024; Xu et al., 2024; Hu et al., 2025b). In the medical domain, not all plausible action combinations can be exhaustively annotated or collected in the training data. Our work addresses this gap by restructuring standard datasets using a reusable split strategy.

## 2.3 CHAIN OF THOUGHT

Chain of Thought (CoT) prompting (Wei et al., 2022) has emerged as an effective strategy to improve the reasoning capabilities of large language models (LLMs). Unlike direct answer generation, CoT encourages models to generate intermediate reasoning steps, thereby enabling better performance on tasks requiring multi-hop inference or arithmetic logic. Subsequent research (Kojima et al., 2022; Li et al., 2025b) explored automated generation of CoT demonstrations, introducing zero-shot CoT prompting without manual exemplars. In the multi-modal domain, multimodal reasoning frameworks such as MuKCoT (Qiu et al., 2024) leverage LLM-generated knowledge-enriched chains of thought to improve knowledge-based VQA. Inspired by these advances, we design a Chain-of-Surgery (CoS) prompting mechanism tailored to surgical scene understanding. CoS simulates a staged diagnostic process—first glancing at a keyframe, then gazing at surrounding temporal context, and finally thinking through a triplet inference. This contrasts with prior works that use flat prompts or retrieve captions independently, enabling more coherent multi-step reasoning in visual understanding tasks.

## 3 METHODOLOGY

The overall framework comprises two synergistic components as shown in Figure 2: a LVLM surgical agent, which is responsible for progressive semantic reasoning, and a SAM-enhanced Machine Encoding, which focuses on capturing spatially and temporally fine-grained visual evidence. The LVLM surgeon is guided by a novel CoS prompting strategy, simulating human-like, stage-wise reasoning over surgical procedures. The machine assistant leverages pretrained encoders to extract spatial region features and video-level temporal dynamics, offering detailed grounding cues to complement the LVLM's reasoning. The outputs of the two parts are then integrated via a lightweight yet expressive fusion block called MiST, which aligns spatial, temporal, and semantic information through attention-based aggregation to enable structured action prediction.

Finally, to evaluate in a fair and extensible way under realistic generalization scenarios, we design a flexible compositional split strategy that converts any existing surgical triplet recognition dataset into a compositional zero-shot benchmark. This ensures that individual components remain seen during training, while their combinations are held out for testing.

## 3.1 SURGICAL REASONING AGENT WITH CHAIN OF SURGERY

LVLMs possess a natural capacity for generalization across previously unseen images, videos, and text, owing to the vast and diverse data they are pre-trained on. However, the sparse and specialized nature of surgical visual data presents unique challenges that cannot be effectively addressed through general-purpose prompting alone. This low prevalence and domain specificity underscore the need for a multi-stage chain-of-surgery prompting paradigm to elicit clinically aligned, context-aware reasoning from the model.

Analogous to the reasoning process of human surgeons, we propose a cognitive-awakening, three-stage analysis pipeline that progressively decouples the spatial and temporal information from surgical video clips, ultimately enabling structured triplet classification.

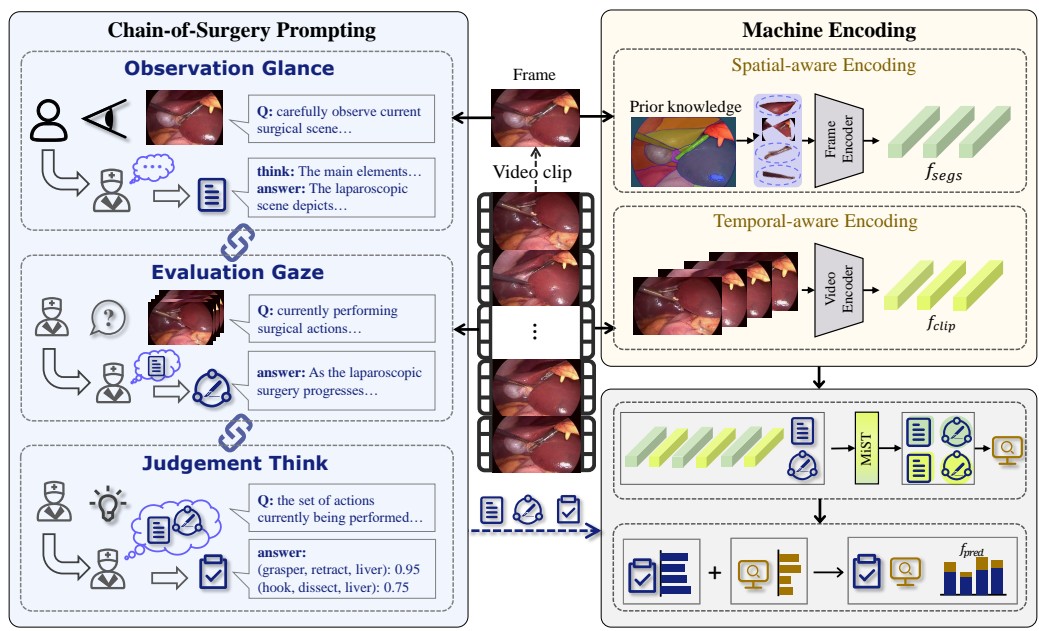

Figure 2: Framework of our CoCoS. We propose the Chain-of-Surgery Prompting (left) to establish cognitive awakening through three stages: Glance (scene-level observation), Gaze (action-level understanding), and Think (structured triplet prediction). Through this process of cognitive awakening, the model is aware of more surgical scene-related context. The spatial and temporal features extracted by Machine Encoding (right) are used in Multimodal image-Sequence-Text (MiST) to reinforce the stability of the whole framework.

In the first stage, termed open observation glance, a vision-language model endowed with visual reasoning capabilities observes a representative key frame from the surgical video in an open-ended manner. This stage is designed to initiate domain-specific reasoning, prompting the agent to adopt a surgery-oriented cognitive mode rather than a generic one. Following this phase, the agent is expected to internalize a clinical perspective in its subsequent reasoning.

In the second stage, called evaluation gaze, the agent is exposed to the entire surgical video clip, allowing it to perceive the temporal dynamics and procedural flow underlying the current operation. This stage emphasizes the recognition of task-relevant cues that emerge over time and are critical for accurate triplet inference.

Finally, in the judgement think stage, the agent performs structured inference by integrating the spatiotemporal context accumulated from the earlier stages. This phase is dedicated to holistic and clinically aligned decision-making, aiming to generate precise triplet predictions grounded in the long-range procedural context.

Compared to the Chain-of-Thought (CoT) paradigm (Figure 3), our Chain-of-Surgery (CoS) introduces a multimodal and clinically grounded reasoning flow. By progressively delivering both visual and textual cues across stages, CoS enables the agent to form a richer internal representation that aligns with human surgical cognition. As shown in Figure 4, the Glance stage primarily focuses on describing the visual scene, while the Gaze stage involves more causal reasoning, helping to resolve uncertainties in the initial description.

## 3.2 MACHINE ENCODING

While LVLMs excel in complex, multi-faceted reasoning tasks, their performance often deteriorates in simpler or lower-level perception tasks due to domain shifts and hallucination. To mitigate this, we incorporate machine encoding modules to provide stable, grounded representations of spatial and

temporal surgical cues. As illustrated in Figure 5, when LVLMs hallucinate in some circumstances, the Machine Encoding module can correct these errors.

### 3.2.1 SPATIAL AND TEMPORAL-AWARE REPRESENTATION.

Given a video clip $X = \{x_1, x_2, ..., x_L\}$, we first apply a segmentation model such as SAM on every frame and get a series of segments as in Eq. 1, where $N_i$ denotes the total number of segments within the i-th frame of the clip:

$$S = \left\{ [s_1^1, s_1^2, \ldots, s_1^{N_1}], \ldots, [s_L^1, s_L^2, \ldots, s_L^{N_L}] \right\}. \tag{1}$$

To filter out valid segments unsupervisedly, i.e., the instruments and the anatomical structures, we manually design prior rules based on domain knowledge such as the area and the continuity of segmentation. As a result, we retain the top-k segmentation with the highest predicted IoU and stability scores, provided that their mask pixel count exceeds a predefined threshold as follows:

$$S' = \left\{ [s_1^1, \ldots, s_1^K, x_1], \ldots, [s_L^1, \ldots, s_L^K, x_L] \right\}, \tag{2}$$

$$f_{\text{segs}} = \text{Enc}_{\text{img}}(S'), \quad f_{\text{clip}} = \text{Enc}_{\text{vid}}(X). \tag{3}$$

After adding the corresponding frame to the segment list as shown in Eq. 2, we feed $S'$ and $X$ into the image encoder and video encoder, respectively, to obtain $f_{\text{segs}} \in \mathbb{R}^{T \times (K+1) \times C' \times H' \times W'}$ and $f_{\text{clip}} \in \mathbb{R}^{C' \times D}$, where $T$ denotes the number of frames within a clip, $K$ is the number of retained segmentations per frame.

In this way, the framework jointly leverages both the fine-grained spatial cues from the segmentation masks and the global temporal dynamics from the video, ensuring that subsequent modules can exploit complementary information from multiple levels of representation.

### 3.2.2 MULTIMODAL IMAGE–SEQUENCE–TEXT FUSION.

To enhance the reasoning ability of the LVLM and mitigate its susceptibility to hallucinations in straightforward or unambiguous surgical scenarios, we propose a dedicated multimodal fusion module, termed MiST. This module is specifically designed to integrate domain-grounded features derived from static segmentation, temporal dynamics, and textual prompts. By consolidating these heterogeneous modalities, MiST facilitates structured multimodal reasoning within the proposed Chain-of-Surgery framework, thereby enabling more reliable and context-aware surgical video understanding.

We first enhance the raw text features via self-attention and project them to a common feature space:

$$F_{\text{text}} = \text{Proj}_{\text{text}}(\text{SelfAtten}(f_{\text{text}})). \tag{4}$$

Next, we use $F_{\text{text}}$ as a query to extract semantically aligned spatial representations. Specifically, we compute instrument-related and target-related features using independent cross-attention blocks, followed by residual fusion:

$$f_{\text{segs}}^{\{\text{I,T}\}} = \text{Proj}_{\{\text{I,T}\}}(f_{\text{segs}}), \tag{5}$$

$$F_{\{\text{I,T}\}} = \text{CrossAtten}_{\{\text{I,T}\}}(F_{\text{text}}, f_{\text{segs}}^{\{\text{I,T}\}}, f_{\text{segs}}^{\{\text{I,T}\}}) + f_{\text{segs}}^{\{\text{I,T}\}}. \tag{6}$$

For the verb component, we align temporal video features to the semantic context using a separate cross-attention block. In this case, the temporal features serve as query tokens, the text as keys, and the projected segmentation as value embeddings:

$$F_{\text{V}} = \text{CrossAtten}_{\text{verb}}(\text{Proj}_{\text{clip}}(f_{\text{clip}}), F_{\text{text}}, \text{Proj}_{\text{seg}}(f_{\text{segs}})). \tag{7}$$

### 3.3 COMPOSITIONAL ZERO-SHOT SURGICAL TRIPLET RECOGNITION

Traditional surgical triplet recognition tasks assume that all candidate triplets appear during training. While effective in limited settings, substantial cases in real-world surgical environments do not satisfy this assumption. To simulate this realistic and more challenging scenario, we propose compositional zero-shot surgical triplet recognition—a new problem formulation aimed at evaluating a model's ability to generalize to unseen compositions of previously observed components.

Let $\mathcal{T} = (i, v, t)$ denote the space of surgical triplets, where each triplet is composed of an instrument $i \in I$, an action verb $v \in V$, and a target $t \in T$. In conventional settings, the training and test sets share the same triplet space $\mathcal{T}_{\text{train}} = \mathcal{T}_{\text{test}}$. In our compositional zero-shot setup, however, we impose the following constraint:

---

**Algorithm 1** Compositional Zero-Shot Split Construction

**Input**: dataset of labeled frames $\mathcal{D}$
**Parameter**: $\gamma$
**Output**: $\mathcal{T}_{\text{train}}, \mathcal{T}_{\text{test}}$

1: Initialize empty count map $C$
2: **for all** frame $(x, (i, v, t))$ in $\mathcal{D}$ **do**
3:     $C[(i, v, t)] \leftarrow C[(i, v, t)] + 1$
4: **end for**
5: $\mathcal{T}_{\text{test}} \leftarrow \{(i, v, t) \mid C[(i, v, t)] < \gamma\}$
6: $\mathcal{T}_{\text{train}} \leftarrow \mathcal{T} \setminus \mathcal{T}_{\text{test}}$
7: **for all** $(i, v, t)$ in $\mathcal{T}_{\text{test}}$ **do**
8:     **if** $i \notin I_{\text{train}}$ **or** $v \notin V_{\text{train}}$ **or** $t \notin T_{\text{train}}$ **then**
9:         Move $(i, v, t)$ from $\mathcal{T}_{\text{test}}$ to $\mathcal{T}_{\text{train}}$
10:     **end if**
11: **end for**
12: **return** $\mathcal{T}_{\text{train}}, \mathcal{T}_{\text{test}}$

---

$$(i, v, t) \in \mathcal{T}_{\text{test}} \quad \text{such that} \quad (i, v, t) \notin \mathcal{T}_{\text{train}}, \tag{8}$$

$$i, v, t \in I_{\text{train}} \times V_{\text{train}} \times T_{\text{train}}. \tag{9}$$

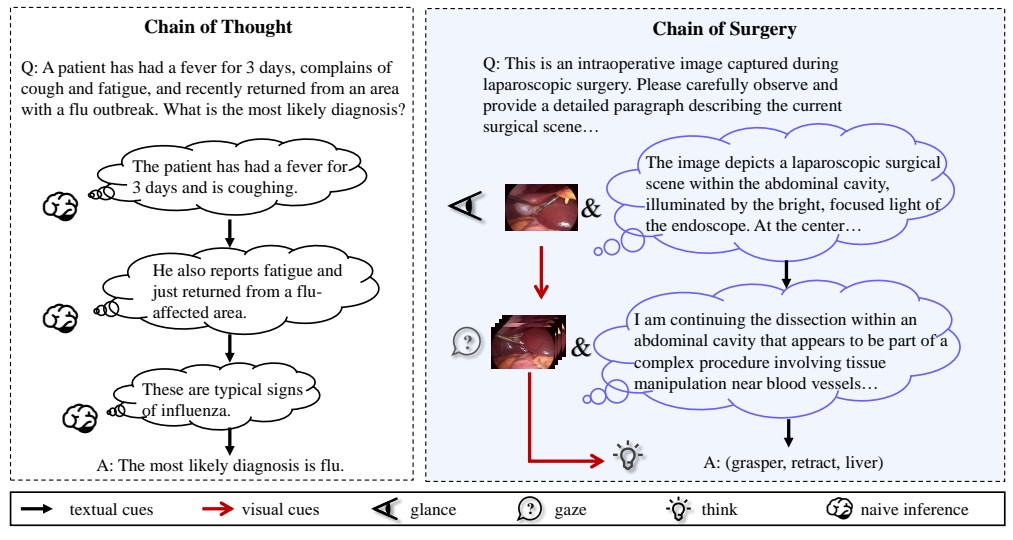

Figure 3: Illustration of the difference between Chain-of-Thought and Chain-of-Surgery reasoning. The left example shows a conventional Chain-of-Thought (CoT) process in a medical diagnosis scenario, where reasoning unfolds solely through textual cues. In contrast, the right example presents a Chain-of-Surgery (CoS) process for intraoperative scene understanding, which progresses through three stages, which are Glance, Gaze, and Think, while explicitly incorporating both textual and visual cues at each stage.

That is, each individual component must appear in the training set, but their composition must not. This encourages models to perform compositional generalization, i.e., the ability to understand novel compositions. Based on this setup, we design principled Algorithm 1 to transform existing surgical triplet recognition datasets into compositional zero-shot splits.

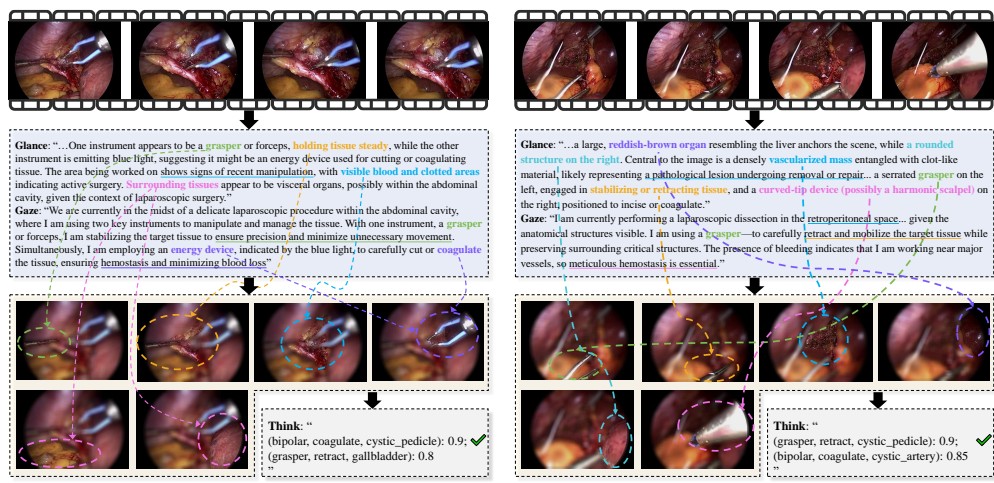

Figure 4: Visualization of textual description generated in glance and gaze. The multi-color lines indicate the corresponding phrases and segments in spatial-aware encoding, and the underlines highlight causal relations associated with the respective phrases.

| Type | Method | $mAP_i$ | $mAP_v$ | $mAP_t$ | $mAP_{ivt}$ |
|---|---|---|---|---|---|
| Conventional | RDV (Nwoye et al., 2022) | 36.90 | 10.84 | 6.91 | 2.52 |
| | RiT (Sharma et al., 2023) | 26.75 | 12.60 | 12.49 | 2.02 |
| VLM | HecVL (Yuan et al., 2024a) | 21.91 | 17.69 | 16.43 | 7.78 |
| | PeskaVLP (Yuan et al., 2024b) | 20.10 | 16.23 | 14.14 | 6.72 |
| | SurgVLP (Yuan et al., 2025) | 21.44 | 18.44 | 15.87 | 5.95 |
| | **Ours (Qwen-VL-Max&QVQ-Max)** | **52.54** | **37.56** | 21.55 | 9.39 |
| | **Ours (Qwen3-VL-8B-Instruct&Thinking)** | 32.82 | 27.67 | **22.30** | **9.98** |

Table 1: Performance comparison on the CholecT50 dataset with the compositional zero-shot setting.

Additionally, to adapt to the compositional zero-shot setup, we redefine the basic recognition unit from a single frame to a short video clip depicting the same action triplet label. Each clip consists of a fixed number of loosely consecutive frames (e.g., 16), all sampled from the same surgical video in their natural chronological order.

## 4 EXPERIMENTS

### 4.1 ZERO-SHOT TRIPLET SPLITTING

Following the data partitioning strategy introduced as Algorithm 1, we construct a compositional zero-shot setting by selecting 40 triplets with less than 80 labeled frames for testing and using the remaining 60 for training. The threshold $\gamma = 80$ ensures a sufficient support for both training and evaluation on CholecT50. This splitting strategy guarantees that all test components are seen during training while enforcing compositional disjointness at the triplet level.

### 4.2 DATASET AND EVALUATION METRICS

We conduct experiments on the CholecT50 dataset, which contains dense frame-level triplet annotations from real-world laparoscopic cholecystectomy videos. To support temporal modeling and ensure semantic consistency, each video is divided into non-overlapping 16-frame clips, where all frames share the same triplet label and preserve their natural chronological order. This preprocessing ensures that each clip is both temporally and semantically coherent. We report mean average

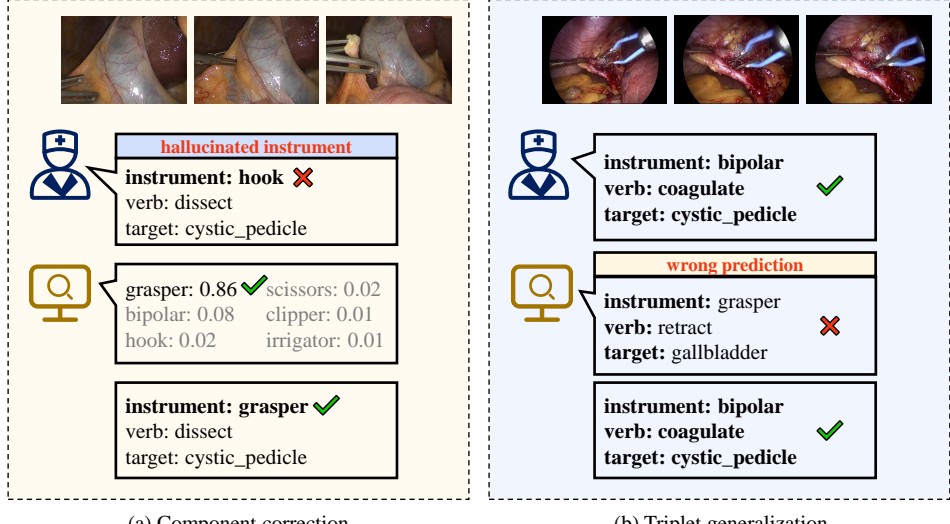

(a) Component correction          (b) Triplet generalization

Figure 5: The case study of (a) component correction and (b) triplet generalization. While the conventional model excels at correcting hallucinated predictions in individual components like instruments, the LVLM shows stronger generalization in predicting unseen full triplets. This highlights their complementary strengths and the necessity of coordinated reasoning in surgical scene understanding.

| Method | mAP$_i$ | mAP$_v$ | mAP$_t$ | mAP$_{ivt}$ |
|---|---|---|---|---|
| ME | 24.06 | 19.34 | 16.36 | 2.50 |
| Glance | 21.20 | 32.90 | 19.57 | 3.90 |
| Glance+Gaze | 38.19 | 34.06 | 18.25 | 6.52 |
| Glance+Gaze+Think | 39.79 | 36.31 | 19.79 | 6.61 |
| Glance+Gaze+Think+ME | 52.54 | 37.56 | 21.55 | 9.39 |

Table 2: Ablation study for Chain-of-Surgery (CoS) with Qwen-VL-Max&QVQ-Max setting.

precision (mAP) of the whole triplet and every single component at the clip level, aligning with the model's prediction granularity.

### 4.3 IMPLEMENTATION DETAILS

Our framework is implemented using PyTorch and trained on a single NVIDIA RTX 3090 GPU. For machine encoding, we employ ResNet-18 and MViT for spatial and temporal encoding, respectively. The textual descriptions generated during inference are encoded using BioBERT before fusing with MiST. For the chain of surgery, we use QVQ-Max for the Glance stage, which conducts open-ended visual observation and summarization of the current surgical scene, and Qwen-VL-Max for the subsequent Gaze and Think stages, which performs deeper reasoning about ongoing surgical actions and full triplet composition. To further prove the power of CoS, we also conduct an extra experiment, which replaces QVQ-Max and Qwen-VL-Max with open-source Qwen3-VL-8B-Thinking and Qwen3-VL-8B-Instruct separately.

### 4.4 RESULTS AND ANALYSIS

We test our method on the CholecT50 dataset under the proposed compositional zero-shot setting, where the model must recognize novel combinations of surgical instruments, actions, and targets. As shown in Table 1, our approach outperforms both traditional visual recognition models and recent VLM-based pretrained methods across all evaluation metrics, including individual triplet components and the full triplet composition.

The most substantial improvement comes in verb prediction, with performance more than doubling compared to the second-best approach. This is typically a shortage for VLMs, especially those built on CLIP, which tend to struggle with temporal reasoning. In contrast, our Chain-of-Surgery framework explicitly constructs temporality-related context with a quick glance at the scene, then gazing on motion cues before thinking out a conclusion. That progression seems to help the model better interpret dynamic patterns of activity.

For components that are mostly spatial, such as instrument and target recognition, our method consistently outperforms existing baselines. Conventional models typically perform well on instruments, likely because these tools have stable and distinctive appearances. Targets, on the other hand, are more challenging due to their greater visual variability and frequent occlusion within the surgical field. By grounding the visual evidence within temporal and textual context, it becomes easier to disambiguate tools from surrounding clutter and to more reliably identify subtle targets like tissue regions or anatomical landmarks. This suggests that even for tasks dominated by static visual cues, the added reasoning and multimodal alignment in our framework can provide a meaningful boost, especially when visual signals alone are incomplete or ambiguous.

In full triplet prediction, our model achieves 9.39 on the $mAP_{ivt}$ metric, which is nearly quadrupling the performance of RDV and outperforming all other VLM baselines. This reflects the challenge of combining correct predictions across all three components and indicates the strength of our multimodal fusion and structured reasoning. As shown in Figure 6, due to the strong generalization ability of CoS, our method achieves particularly notable improvements in several challenging triplet categories compared to HecVL (Yuan et al., 2024a) which performs second-best in $mAP_{ivt}$.

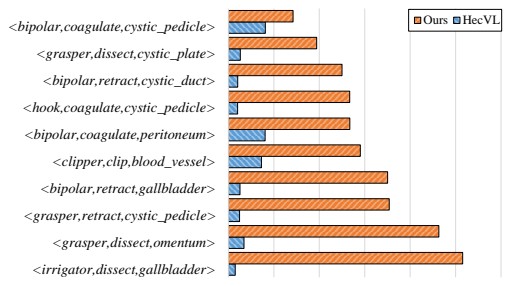

Figure 6: Top-10 triplets improvements in $mAP_{ivt}$ of our framework compared to the HecVL (the second-best method).

### 4.5 ABLATION STUDY

We conduct an ablation study to validate the effectiveness of Chain-of-Surgery. Table 2 shows the individual contributions of each stage in our Chain-of-Surgery (CoS) prompting. Starting from the base case of Glance, which performs an initial scene description from a single frame, we observe limited performance across all triplet components, especially in full triplet prediction. After adding the Gaze stage, which introduces temporal dynamics by prompting based on short clips, brings a noticeable improvement. Finally, incorporating the Think stage yields further gains, especially in verb recognition, by improving $mAP_{ivt}$ by 3.41 compared to the Glance stage.

When compared with the performance of the whole framework as shown in Table 1, we find the complete framework yields a substantial jump to 9.39 $mAP_{ivt}$. This gap underscores the complementary role of our spatial-temporal visual encoding pipeline, which inherits robustness from conventional models and grounds CoS reasoning more effectively.

### 5 CONCLUSION AND DISCUSSION

In this paper, we propose a novel Cognitive-Awakening Chain-of-Surgery (CoCoS) framework to tackle the more challenging yet realistic task of compositional zero-shot surgical triplet recognition. This design bridges the strengths of both conventional encoders and modern LVLMs. Experimental results demonstrate that this staged reasoning approach makes LVLMs not only more interpretable but also more reliable. The Glance–Gaze–Think pattern underpinning Chain-of-Surgery (CoS) mirrors how humans perceive and interpret complex, dynamic surgical scenes and guides the model on when to observe, when to focus, and when to conclude.

Future work will focus on narrowing the remaining gap between seen and unseen triplet compositions, which remains a key challenge in compositional generalization. Additionally, modeling more adaptive temporal reasoning and enhancing real-time applicability in real-world scenarios are also important directions for future optimization.

## 6 ETHICS STATEMENT

This study uses only publicly available endoscopic video datasets with all patient-identifiable information removed. No additional human or animal data were collected, and no personally identifiable information is involved. We do not foresee any ethical concerns in this work.

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

# A APPENDIX

## A.1 USE OF LARGE LANGUAGE MODELS (LLMS)

We used large language models (LLMs), specifically OpenAI's ChatGPT (GPT-4o/5), as an assistive tool during the preparation of this paper. The LLMs were used for language polishing, grammar refinement, and improving readability of the text. All technical content, data interpretation, and scientific contributions were generated entirely by the authors. The authors take full responsibility for the correctness, originality, and integrity of the content presented in this paper.

