# Cognitive-Awakening Chain-of-Surgery for Compositional Zero-Shot Surgical Triplet Recognition (Supplementary)

## 1 Triplet Category Splitting

We use $\gamma = 80$ in the paper, which is a suitable setting for both training and evaluation on CholecT50. As shown in Table 4, a total of 60 triplet types are used for training, with 40 compositional-unseen categories held out for evaluating and testing.

## 2 Chain-of-Surgery Prompting Text

The prompts used in CoS are shown in Figure 4. In the gaze stage, thinking contents are concatenated with the answers generated from the glance stage directly as the assistant prompt. In the think stage, answers generated from the gaze stage are used as assistant prompts separately.

## 3 Implementation Details

We adopt the AdamW optimizer with a learning rate of 0.001 and a weight decay of 0.0001. Binary Cross-Entropy loss is used for instrument, verb, target, and triplet recognition, while an additional InfoNCE loss is applied between the text representations produced at the Gaze stage and the Video Encoder features. The batch size is set to 4, corresponding to four video clips per batch. Both the Frame Encoder and Video Encoder are trained with an early-stopping strategy, using a patience of 5 and a minimum delta of 0.005.

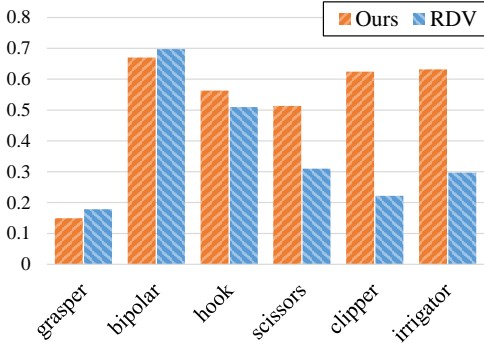

Figure 1: Instrument improvements in $\text{mAP}_i$ of our framework compared to the RDV (the second-best method).

| Method | $mAP_i$ | $mAP_v$ | $mAP_t$ | $mAP_{ivt}$ |
|---|---|---|---|---|
| Glance | 13.62 | 15.54 | 9.24 | 2.11 |
| Glance+Gaze | 19.07 | 17.85 | 9.47 | 2.35 |
| Glance+Gaze+Think | 19.88 | 19.26 | 10.52 | 2.51 |
| Glance+Gaze+Think+Machine Encoding | 32.82 | 27.67 | 22.30 | 9.98 |

Table 1: Ablation study for Chain-of-Surgery (CoS) with Qwen3-VL-8B-Instruct&Qwen3-VL-8B-Thinking setting.

| clip length $T$ | $mAP_i$ | $mAP_v$ | $mAP_t$ | $mAP_{ivt}$ |
|---|---|---|---|---|
| 8 | 49.70 | 32.36 | 20.15 | 3.31 |
| 16 | 52.54 | 37.56 | 21.55 | 9.39 |
| 32 | 51.41 | 37.14 | 20.72 | 9.26 |

Table 2: Ablation study for clip length $T$

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

---

**Algorithm 1** SAM-based Region Selection Pipeline

---

1: **Input:** Image $I$, top-$k$
2: **Output:** Selected masks
3: $\mathcal{S} \leftarrow$ SELECTREGIONS$(I, k)$
4: **return** $\mathcal{S}$

5: **function** SELECTREGIONS$(I, k)$
6: $\quad \mathcal{M} \leftarrow$ SamAutomaticMaskGenerator$(I)$
7: $\quad \mathcal{M} \leftarrow \{m \in \mathcal{M} \mid \text{area}(m) \geq 2000\}$
8: $\quad \mathcal{F} \leftarrow \emptyset$
9: $\quad$ **for all** $m \in \mathcal{M}$ **do**
10: $\quad\quad$ **if** ISCONTINUOUS$(m)$ **then**
11: $\quad\quad\quad \mathcal{F} \leftarrow \mathcal{F} \cup \{m\}$
12: $\quad\quad$ **end if**
13: $\quad$ **end for**
14: $\quad \mathcal{R} \leftarrow$ SortByScore$(\mathcal{F})$
15: $\quad$ **return** first $k$ masks in $\mathcal{R}$
16: **end function**

17: **function** ISCONTINUOUS$(mask)$
18: $\quad r \leftarrow$ CHECKROWCONTINUITY$(mask)$
19: $\quad c \leftarrow$ CHECKCOLUMNCONTINUITY$(mask)$ $\qquad\qquad\qquad\qquad \triangleright r, c \in \{\text{True}, \text{False}\}$
20: $\quad$ **if** $r = \text{False}$ **and** $c = \text{False}$ **then**
21: $\quad\quad$ **return** False $\qquad\qquad\qquad \triangleright$ Both directions broken $\rightarrow$ discard mask
22: $\quad$ **else**
23: $\quad\quad$ **return** True $\qquad\qquad \triangleright$ At least one direction is continuous $\rightarrow$ keep
24: $\quad$ **end if**
25: **end function**

---

| Type | Method | mAP$_i$ | mAP$_v$ | mAP$_t$ | mAP$_{ivt}$ |
|------|--------|---------|---------|---------|-------------|
| Conventional | RDV (Nwoye et al., 2022) | **34.24** | 10.77 | 7.16 | 2.80 |
| | RiT (Sharma et al., 2023) | 27.29 | 13.08 | 13.42 | 3.25 |
| VLM | HecVL (Yuan et al., 2024a) | 16.43 | 14.37 | 14.08 | 6.61 |
| | PeskaVLP (Yuan et al., 2024b) | 17.62 | 13.62 | 14.03 | 6.88 |
| | SurgVLP (Yuan et al., 2025) | 15.29 | 15.03 | 15.01 | 5.52 |
| | **Ours (Qwen3-VL-8B-Instruct&Thinking)** | 31.55 | **26.72** | **24.11** | **9.56** |

Table 3: Performance comparison on the ProstaTD(Chen et al., 2025) dataset with the compositional zero-shot setting.

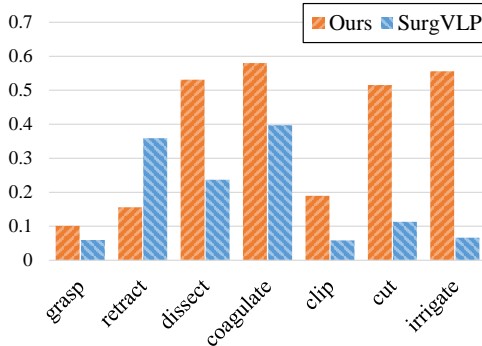

Figure 2: Verb improvements in mAP$_v$ of our framework compared to the SurgVLP (the second-best method).

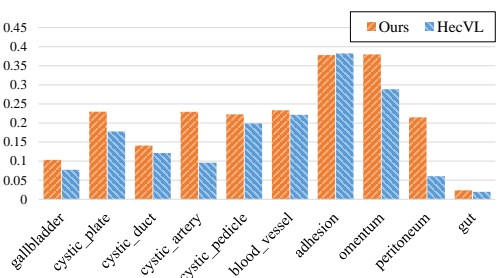

Figure 3: Target improvements in mAP$_t$ of our framework compared to the HecVL (the second-best method).

---

**Glance**

User Prompt:
This is an intraoperative image captured during laparoscopic surgery.
Please carefully observe and provide a detailed paragraph describing the current surgical scene, including the anatomical structures, surgical instruments, and ongoing procedures visible in the image.

---

**Gaze**

System Prompt:
You are a highly experienced laparoscopic surgeon.

Assistant Prompt:
[think from glance] + [answer from glance]

User Prompt:
You are provided with a video clip composed of multiple consecutive frames from the same laparoscopic surgery as the previous image.
As the operating surgeon, describe in one paragraph what surgical actions you are currently performing, referring to both the ongoing procedure and any anatomical or procedural context you can observe.

---

**Think**

System Prompt:
You are a professional laparoscopic surgeon and an expert in surgical scene annotation.
Your task is to predict surgical actions based on intraoperative images, strictly following a structured format.

Assistant Prompt:
[think from glance] + [answer from glance]
[answer from gaze]

User Prompt:
This is an intraoperative image from the same laparoscopic surgery as before.
Based on this image, predict the set of actions currently being performed.
**Important Instructions**:
Each action must be represented as a structured triplet with the following fields:
  - instrument: must be chosen from the following list: ['grasper', 'bipolar', 'hook', 'scissors', 'clipper', 'irrigator'].
  - verb: must be chosen from the following list: ['grasp', 'retract', 'dissect', 'coagulate', 'clip', 'cut', 'aspirate', 'irrigate', 'pack', 'null_verb'].
  - target: must be chosen from the following list: ['gallbladder', 'cystic_plate', 'cystic_duct', 'cystic_artery', 'cystic_pedicle', 'blood_vessel', 'fluid', 'abdominal_wall_cavity', 'liver', 'adhesion', 'omentum', 'peritoneum', 'gut', 'specimen_bag', 'null_target'].
  - confidence: a float number between 0 and 1 indicating your confidence in the prediction.
Only select values from the given lists. Do not invent or hallucinate new labels.
If you are uncertain, select the most plausible option based on visual evidence.
**Output Format**: Output only a JSON-formatted list of triplets. Do not include any explanations, commentary, or text.

---

Figure 4: Text prompts used in the Chain-of-Surgery (CoS).

| triplet category for validation | | triplet category for training | | | |
|---|---|---|---|---|---|
| id | name | id | name | id | name |
| 0 | <grasper, dissect, cystic_plate> | 1 | <grasper, dissect, gallbladder> | 52 | <hook, coagulate, liver> |
| 2 | <grasper, dissect, omentum> | 4 | <grasper, grasp, cystic_duct> | 55 | <hook, cut, peritoneum> |
| 3 | <grasper, grasp, cystic_artery> | 6 | <grasper, grasp, cystic_plate> | 57 | <hook, dissect, cystic_artery> |
| 5 | <grasper, grasp, cystic_pedicle> | 7 | <grasper, grasp, gallbladder> | 58 | <hook, dissect, cystic_duct> |
| 8 | <grasper, grasp, gut> | 9 | <grasper, grasp, liver> | 59 | <hook, dissect, cystic_plate> |
| 15 | <grasper, retract, cystic_pedicle> | 10 | <grasper, grasp, omentum> | 60 | <hook, dissect, gallbladder> |
| 24 | <bipolar, coagulate, cystic_artery> | 11 | <grasper, grasp, peritoneum> | 61 | <hook, dissect, omentum> |
| 25 | <bipolar, coagulate, cystic_duct> | 12 | <grasper, grasp, specimen_bag> | 62 | <hook, dissect, peritoneum> |
| 26 | <bipolar, coagulate, cystic_pedicle> | 13 | <grasper, pack, gallbladder> | 63 | <hook, retract, gallbladder> |
| 31 | <bipolar, coagulate, peritoneum> | 14 | <grasper, retract, cystic_duct> | 64 | <hook, retract, liver> |
| 32 | <bipolar, dissect, adhesion> | 16 | <grasper, retract, cystic_plate> | 66 | <scissors, cut, adhesion> |
| 35 | <bipolar, dissect, cystic_plate> | 17 | <grasper, retract, gallbladder> | 68 | <scissors, cut, cystic_artery> |
| 38 | <bipolar, grasp, cystic_plate> | 18 | <grasper, retract, gut> | 69 | <scissors, cut, cystic_duct> |
| 41 | <bipolar, retract, cystic_duct> | 19 | <grasper, retract, liver> | 71 | <scissors, cut, liver> |
| 42 | <bipolar, retract, cystic_pedicle> | 20 | <grasper, retract, omentum> | 76 | <scissors, dissect, omentum> |
| 43 | <bipolar, retract, gallbladder> | 21 | <grasper, retract, peritoneum> | 78 | <clipper, clip, cystic_artery> |
| 45 | <bipolar, retract, omentum> | 22 | <bipolar, coagulate, abdominal_wall_cavity> | 79 | <clipper, clip, cystic_duct> |
| 46 | <hook, coagulate, blood_vessel> | 23 | <bipolar, coagulate, blood_vessel> | 82 | <irrigator, aspirate, fluid> |
| 47 | <hook, coagulate, cystic_artery> | 27 | <bipolar, coagulate, cystic_plate> | 84 | <irrigator, dissect, cystic_pedicle> |
| 48 | <hook, coagulate, cystic_duct> | 28 | <bipolar, coagulate, gallbladder> | 87 | <irrigator, dissect, omentum> |
| 49 | <hook, coagulate, cystic_pedicle> | 29 | <bipolar, coagulate, liver> | 88 | <irrigator, irrigate, abdominal_wall_cavity> |
| 50 | <hook, coagulate, cystic_plate> | 30 | <bipolar, coagulate, omentum> | 90 | <irrigator, irrigate, liver> |
| 53 | <hook, coagulate, omentum> | 33 | <bipolar, dissect, cystic_artery> | 92 | <irrigator, retract, liver> |
| 54 | <hook, cut, blood_vessel> | 34 | <bipolar, dissect, cystic_duct> | 93 | <irrigator, retract, omentum> |
| 56 | <hook, dissect, blood_vessel> | 36 | <bipolar, dissect, gallbladder> | 94 | <grasper, null_verb, null_target> |
| 65 | <scissors, coagulate, omentum> | 37 | <bipolar, dissect, omentum> | 95 | <bipolar, null_verb, null_target> |
| 67 | <scissors, cut, blood_vessel> | 39 | <bipolar, grasp, liver> | 96 | <hook, null_verb, null_target> |
| 70 | <scissors, cut, cystic_plate> | 40 | <bipolar, grasp, specimen_bag> | 97 | <scissors, null_verb, null_target> |
| 72 | <scissors, cut, omentum> | 44 | <bipolar, retract, liver> | 98 | <clipper, null_verb, null_target> |
| 73 | <scissors, cut, peritoneum> | 51 | <hook, coagulate, gallbladder> | 99 | <irrigator, null_verb, null_target> |
| 74 | <scissors, dissect, cystic_plate> | | | | |
| 75 | <scissors, dissect, gallbladder> | | | | |
| 77 | <clipper, clip, blood_vessel> | | | | |
| 80 | <clipper, clip, cystic_pedicle> | | | | |
| 81 | <clipper, clip, cystic_plate> | | | | |
| 83 | <irrigator, dissect, cystic_duct> | | | | |
| 85 | <irrigator, dissect, cystic_plate> | | | | |
| 86 | <irrigator, dissect, gallbladder> | | | | |
| 89 | <irrigator, irrigate, cystic_pedicle> | | | | |
| 91 | <irrigator, retract, gallbladder> | | | | |

Table 4: Triplet categories splitting list with $\gamma = 80$.