# OpenReview forum: "Cognitive-Awakening Chain-of-Surgery for Compositional Zero-Shot Surgical Triplet Recognition"
_ICLR.cc/2026/Conference — ICLR 2026 Conference Withdrawn Submission_

### Official Review · Reviewer_bsRt · 2025-10-15

**Soundness:** 2
**Presentation:** 2
**Contribution:** 2
**Rating:** 6
**Confidence:** 4

**Summary:**

This paper introduces the "Cognitive-awakening Chain-of-Surgery" (CoCoS) framework to tackle Compositional Zero-Shot Surgical Triplet Recognition (CZSTR), a novel and highly significant research topic for surgical AI generalization. While the problem itself is a valuable contribution, the paper's technical contribution is limited, as the method primarily combines existing models (LVLMs, SAM, MViT) with a new prompting strategy. The work's primary weaknesses are its insufficient experimental validation and poor reproducibility. The ablation study is critically incomplete. Furthermore, crucial implementation details such as hyperparameters and manual filtering rules are omitted, and no code has been released to compensate.

**Strengths:**

1. The paper introduces a novel and challenging task. It addresses a realistic gap by requiring models to recognize unseen combinations of known components, which is crucial for developing truly generalizable and trustworthy surgical AI systems.
2. It creatively leverages VLMs through the proposed "Cognitive-awakening Chain-of-Surgery" (CoCoS) framework. This method simulates a surgeon's cognitive process in three stages (Glance, Gaze, Think) to guide the model toward a deeper, context-aware understanding of complex surgical scenes.

**Weaknesses:**

1. The core idea of adapting a "Chain-of-Thought" reasoning process is not entirely novel in this domain. Even if the proposed 'Chain-of-Surgery' concept differs in its specifics, the authors should cite and discuss the relevant prior work, "Chain-of-look prompting for verb-centric surgical triplet recognition in endoscopic videos."
2. Regarding time efficiency, the use of both SAM, MViT and VLMs must result in a speed trade-off. However, real-time performance is critical in surgery. The paper should report what the exact inference time is, compare it against RDV, and quantify how much speed was sacrificed by the proposed method.
3. Reproducibility is a major concern. It took me a long time to find that 'T' (defined in Line 236) is 16 (in Line 360). The 'Implementation Details' section is almost empty; it has no hyperparameters, and the code has not been released.
4. In section 3.2.1, the authors mention using "manually design prior rules based on domain knowledge." However, these specific rules could not be found in the appendix. Furthermore, based on my knowledge, relying on such rules is unreliable. The size of tools and targets varies significantly with endoscopic zoom or their position at the periphery, meaning they can appear either very large or very small.  Also, since CholecT50 is annotated at 1 fps, it is normal for tools not to appear continuously, which contradicts the rule.
5. Using only one dataset for surgical triplets makes the experiment seem less reliable and limits the findings to the single task of cholecystectomy. Based on my search, the "prostatd: a large-scale multi-source dataset for structured surgical triplet detection" also provides triplet labels and could have been used.

**Questions:**

1.  Based on your Fig. 4, it looks like neither 'glance' nor 'gaze' is of much help for target recognition. Some of the associated phrases for the target seem inaccurate, and the link between the text and the visual target appears tenuous or forced. The results in Table 2 seem to reflect this phenomenon as well.
2.  The choice of 16 as the clip length is not justified. Surgical actions can be long or short, having variable durations. If this value was determined empirically, where is the ablation study to support it?
3.  In Table 2, what are the results for using only the 'Machine Encoding' module? Because the final judgment is based on the combination of both components' predictions, this baseline is essential to understand their individual contributions.
4.   Although the model is designed for unseen (rare) triplets (γ=80), it clearly also has the ability to recognize common triplets in a zero-shot manner. Why not use more diverse splits to test its capabilities?

---

> ### Author Response · Authors · 2025-11-25
> **Reply to Reviewer bsRt**
>
> We thank the reviewer for highlighting the need for stronger experimental validation.
>
> **Q1: Citation Completion**
>
> **A1:** Thank you for pointing out the missing citation.
> We have cited *Chain-of-Look prompting for verb-centric surgical triplet recognition*.
>
> The key distinction between CoCoS and previous CoT lies in its natural three-stage structure of surgical cognition. Each stage addresses a specific information bottleneck—ranging from global context and temporal ambiguities to structured semantic grounding, enhancing visual alignment compared with standard CoT approaches.
>
> **Q2: Inference Time**
>
> **A2:** We appreciate the reviewer's concern regarding inference speed. We would like to clarify that CoCoS is _not_ designed for real-time intraoperative decision-making but rather for *offline postoperative analysis and workflow assessment*.
>
> In detail, **SAM contributes ~85% of the total inference time**, while the **LVLM calls (Qwen-VL-Max / QVQ-Max)** through the API account for most of the remaining overhead. The MiST fusion and Machine Encoding (ResNet18 + MViT) add less than 5% overhead.
>
> Lightweight CoCoS are indeed part of our future work, and we will add a discussion of this in the Conclusion section.
>
> **Q3: Hyperparameters Claim and Code Release**
>
> **A3:** We apologize for the incomplete appendix. We have updated the optimizer configuration, LR, batch size, training strategy, and masks filtering strategy **in the supplementary**. Code will be released after the paper is accepted.
>
> **Q4: Explanation on Manually Designed Prior Rules**
>
> **A4:** Thank you for pointing out the lack of methodological clarity in the paper. The term "continuity" in the paper specifically refers to the spatial continuity of pixels within a single frame, which means that any mask presenting discontinuous pixels along both axes is removed.  **The pseudo of the selection algorithm has been updated to supplementary.**
>
> **Q5: Dataset Generalization**
>
> **A5:** We appreciate the reviewer's concern regarding dataset diversity and generalization. At the time of our initial experiment, CholecT50 was the only publicly accessible dataset that provided large-scale, high-quality triplet annotations,.
>
> In September of this year, the **ProstaTD** dataset [ref1] was released, offering structured surgical triplet annotations. We performed an additional experiment on ProstaTD. The results are presented as **Table 2 in the supplementary**.
>
> This evaluation demonstrates that:
> - CoCoS is not restricted to CholecT50
> - The proposed reasoning framework exhibits strong cross-domain generalizability, transferring effectively from cholecystectomy to prostate surgery.
>
> > Ref:
> > [1] ProstaTD: A Large-scale Multi-source Dataset for Structured Surgical Triplet Detection
>
> **Q6: Hardness in Target Recognition**
>
> **A6:** Target Recognition has always been the most challenging subtask among single component recognitions, owing to issues including ambiguous anatomical borders and bleeding. To overcome these difficulties, rubust fine-grained spatial understanding capability is required. LVLMs are particularly sensitive to such noise and therefore struggle to extract reliable fine-grained spatial cues from global (Glance) or temporal (Gaze) reasoning.
>
> This explains why the direct gain of Glance/Gaze on targets is small, while the combined system (CoS + MiST + Machine Encoding) yields the highest target performance in Table 2.
>
> **Q7: Clip Length of 16**
>
> **A7:** We thank the reviewer for pointing out the missing justification of the clip length. We indeed evaluated different temporal spans ($T=8,16,32$) during the early stage of the project. The ablation results are represented as Table 2 in the supplementary.
>
> **Q8: Machine Encoding Module Only Result**
>
> **A8:** Thank you for pointing out the incomplete ablation result; the result of **"Machine Encoding only"** is added to Table 2 in the paper.
>
> **Q9: Evaluation on Frequent Triplets**
> **A9:** We appreciate the reviewer's recommendation to assess our approach on seen triplet classes. In response, we maintain the original split for unseen triplets and, additionally, **allocate 20% of the samples from each seen triplet class to the test set**. With Qwen-VL-Max + QVQ-Max CoCoS, we report the following results:
>
> | Splitting| $\text{mAP}_i$ | $\text{mAP}_v$ | $\text{mAP}_t$ | $\text{mAP}_{ivt}$ |
> | ------ | :-----: | :----: | :-----: | :-----: |
> | unseen triplet only |52.54|37.56|21.55|9.39|
> | seen&unseen triplet |76.27|47.90|21.31|10.98|
>
> We find that incorporating a subset of seen triplets into the test set results in an overall performance gain, suggesting that **CoCoS maintains strong performance on seen triplets** without exhibiting any degradation.
>
> Paper and supplementary have been updated.

---

> > ### Comment · Reviewer_bsRt · 2025-11-26
> >
> > Thanks for respecting my review. My initial high score was driven by the genuinely interesting and important task the authors propose. I hope the authors keep pushing in this direction. I'll keep my positive recommendation.

---

### Official Review · Reviewer_6Wjd · 2025-10-23

**Soundness:** 2
**Presentation:** 1
**Contribution:** 2
**Rating:** 2
**Confidence:** 4

**Summary:**

The paper proposes CoCoS (Cognitive-awakening Chain-of-Surgery): a staged prompting scheme that queries large VLMs and fuses their text with spatial (ResNet-18+SAM) and temporal (MViT) features via a MiST fusion module to predict surgical triplets <instrument, verb, target> under a compositional zero-shot (CZSL) split of CholecT50. Results report gains over prior triplet models and recent VLM pretraining baselines.

**Strengths:**

- Important problem: zero-shot compositional triplets are clinically relevant for generalization.

**Weaknesses:**

- Baselines are potentially not all capable of composing unseen triplets (e.g., RDV) we makes some of the comparisons quite unfair
- There seems to be a fundamental capacity mismatch between the models used for comparisons, while some use tiny ResNet-18, others essentially do a mixture of powerful foundation models such as Qwen-VL-Max and QVQ-Max with extra encoders
- The approach mainly seems to benefit from the combination of out of the shelf models with limited finetuning.There is a lot of complexity in the method and I don’t think it is necessarily justified. Potentially a much simpler approach that still benefits from Qwen-VL-MAX and QVQ-Max would perform similarly. Generally the ablations don’t isolate the claimed contribution of this paper such as CoS well.
- Training is unclear: which modules are trained, MiST only?
- Under-specified SAM details and reporting: mask prompting/selection, instrument–anatomy mapping etc.

**Questions:**

- Trainable modules & loss: Exactly which components are trained/frozen? What are the objectives for I/V/T and full triplet scoring?
- It would be good to add a split where test triplets are frequent (not just rare) to show true compositionality beyond long-tail effects.
- Add a machine-only baseline: ResNet-18 \+ MViT, no LVLM text, I/V/T heads \+ optional compatibility.
- Capacity-matched comparison: Either replace Qwen-VL-Max/QVQ-Max with an open model of comparable capacity (e.g., Qwen2-VL) or upgrade baselines to similar parameter/compute budgets.

---

> ### Author Response · Authors · 2025-11-25
> **Reply to Reviewer 6Wjd**
>
> We thank the reviewer for highlighting both the novelty of the problem and the need for stronger experimental validation.
>
> **Q1: Fair Comparisons to Baselines**
>
> **A1:** We thank the reviewer for pointing out the concern regarding whether all baselines are capable of composing unseen triplets.
>
> We would like to clarify that **Compositional Zero-Shot Surgical Triplet Recognition (CZSTR)** is a new setting introduced in this work to better reflect real surgical generalization, where `<instrument, verb, target>` combinations. As CZSTR did not previously exist, no conventional triplet recognition methods were specifically designed for this setting. We therefore evaluate representative baselines under the same compositional splits to examine how well existing approaches generalize when faced with unseen triplets.
>
> In contrast, our framework is intentionally designed to combine these two strengths:
>
> - **retain the robustness of conventional encoders** for challenging surgical scenes, and
> - **leverage LVLMs for stronger compositional generalization**, enabling better performance on unseen triplets.
>
> **Q2: Fundamental Models Capacity**
>
> **A2:** The primary objective of CoS is to elicit the latent capabilities of LVLMs in surgical scenarios **without introducing any additional finetuning on LVLM**, thereby enabling these increasingly capable models to be fully exploited in this uncommon domain. As shown in Table 2, the LVLM's performance exhibits a consistent improvement under the stepwise Glance–Gaze–Think prompting strategy.
>
> To demonstrate this, we conducted experiments on the open-source **Qwen3-VL-8B** model. We replace QVQ-Max and Qwen-VL-Max with the Qwen3-VL-8B-Thinking and Qwen3-VL-8B-Instruct separately. Performance comparison results, along with corresponding ablations, limited to the length of comments, are **updated as Table 1 in the paper and Table1 in the supplementary**.
>
> These results show that **even with a smaller open-source LVLM (Qwen3-VL-8B), CoS effectively stimulates its reasoning capability** for our compositional zero-shot triplet recognition task. The ablation demonstrates that each reasoning stage contributes incrementally, culminating in the full performance gain when combined with Machine Encoding.
>
> **Q3: Trainable Modules Clarification**
>
> **A3:** Within the overall pipeline, the components that require training are the Frame Encoder, the Video Encoder, and MiST. Among them, the Frame Encoder and Video Encoder are trained with an early-stopping strategy, whereas MiST continues training throughout. Regarding the loss functions, BCE loss is applied to the instrument, verb, target, and triplet predictions, and an additional InfoNCE loss is introduced between the text and video embeddings. **Details have been updated in the supplementary.**
>
> **Q4: Detailed Explanation on SAM Usage**
>
> **A4:** SAM is used to extract the masks in every frame in the video clip. The **selection algorithm** executes in the following steps:
>
> 1. Generate global masks for an entire image.
> 2. Considering that instruments and anatomical structures usually occupy relatively large regions, all masks with very small areas (e.g., <2,000 pixels) are removed.
> 3. Since reliable masks of instruments and anatomical structures should maintain pixel continuity, we perform row-wise and column-wise scans on each mask. Any mask presenting discontinuous pixels along both axes is removed.
> 4. Retain the top-k segmentation with the highest predicted IoU and stability scores.
>
> **The pseudo of the selection algorithm has been updated to supplementary.**
>
> **Q5: Evaluation on Frequent Triplets**
>
> **A5:** We thank the reviewer for the suggestion to evaluate our method on seen triplet classes as well.
>
> To address this, we keep the original unseen-triplet split unchanged and additionally sample 20% **of each seen triplet class** into the test set, leaving 80% for training. Using Qwen-VL-Max + QVQ-Max CoCoS, we obtain the following results:
>
> | Splitting| $\text{mAP}_i$ | $\text{mAP}_v$ | $\text{mAP}_t$ | $\text{mAP}_{ivt}$ |
> | ----- | :-----: | :------: | :-----: | :-----: |
> | unseen triplet only |52.54|37.56|21.55|9.39|
> | seen&unseen triplet |76.27|47.90|21.31|10.98|
>
> We observe that including a portion of seen triplets in the test set leads to an overall performance improvement, indicating that **CoCoS also performs well on seen triplets** and does *not* suffer from performance degradation.
> **Q6: Machine-only Baseline**
>
> **A6:** Thank you for the suggestion regarding the "machine-encoding-only" comparison. We therefore conducted an experiment in which the CoS component was removed from the framework while retaining the Machine Encoding module:
>
> | Method| $\text{mAP}_i$ | $\text{mAP}_v$ | $\text{mAP}_t$ | $\text{mAP}_{ivt}$ |
> | ---- | :-----: | :-----: | :-------: | :-----: |
> | CoS|39.79|36.31|19.79|3.90|
> | Machine Encoding|24.06|19.34|16.36|2.50|
> | CoS+Machine Encoding |52.54|37.56|21.55|9.39|
>
> Paper and supplementary have been updated.

---

> > ### Comment · Reviewer_6Wjd · 2025-11-25
> >
> > I thank the authors, as they have clearly spend a lot of time on the rebuttal, including new experiments and explanations. They have resolved most of my earlier questions regarding what is trained, how SAM is used etc.
> >
> > However, even after considering these additions, my main concerns remain. At a high level, the method is essentially a carefully engineered pipeline that is designed to benefit from strong LVLMs. The performance is valuable as a proof-of-concept that foundation models, when prompted and orchestrated well, can be leveraged for compositional surgical triplet recognition. Yet, from a machine learning perspective, the core novelty remains limited, there is no fundamentally new learning principle, objective, or architectural ingredient relevant to the community.
> >
> > Overall, I see this work as a solid and interesting engineering effort that illustrates how rapidly improving foundation models can be repurposed for surgical understanding with some task-specific design. However, in my view, this falls short of the high bar of acceptance for ICLR. I therefore retain my original score.

---

### Official Review · Reviewer_fhrs · 2025-10-28

**Soundness:** 2
**Presentation:** 2
**Contribution:** 2
**Rating:** 2
**Confidence:** 3

**Summary:**

The paper tackles compositional zero-shot surgical triplet recognition, predicting <instrument, verb, target> combinations that never co-occurred during training. It proposes a chain-of-surgery pipeline (glance -> gaze -> think) that stages an LVLM's perception and reasoning, then fuses this with a machine encoding branch that supplies grounded spatial and temporal cues. This paper also proposes MiST fusion module aligns text prompts and dynamics via tailored cross-attention.

**Strengths:**

1. Introducing compositional zero-shot recognition for surgical triplets is timely and realistic.
2. A clear, human-inspired scheduling of perception and reasoning for LVLMs.
3. On CholecT50 CZSL, the method outperforms prior VLM baselines; the cumulative gains in the ablation (mAP_ivt from 3.90→9.39) substantiate the contribution of each component.

**Weaknesses:**

1. The Chain-of-Surgery is compelling, but the paper should articulate more sharply how its staged prompting differs in principle from existing multi-step/multi-view LVLM prompting beyond domain adaptation (e.g., why exactly three stages; what information bottlenecks each stage resolves).
2. THis paper argues that there are 3 contributions. But I think the first two are actually one contribution. Moreover, the proposed chain-of-surgery is no big difference with former methods and has no contribution for the community. The improvement on performance comes from the tuning process and not the prompt.
3. Why you design MiST? what is the purpose of it? And lack the corresponding ablation studies.

**Questions:**

What is the model scale used in your model and the compared methods?

---

> ### Author Response · Authors · 2025-11-25
> **Reply to Reviewer fhrs**
>
> We thank the reviewer for the constructive feedback. We address the concerns on contribution clarity and MiST motivation as follows.
>
> **Q1: Distinction of CoS over Regular Multi-Step Prompting**
>
> **A1:** Thank you for pointing this out. We agree our exposition could better articulate the principled reasoning behind the three stages.
>
> Key difference from existing multi-step prompting:
>
> 1. Surgical cognition has a natural three-stage structure **with visual cues throughout**
> 	- Glance: global situational assessment
> 	- Gaze: temporal process understanding
> 	- Think: structured semantic integration
>
> 2. Each stage resolves a different information bottleneck
>     - Glance: overcomes missing global cues and ambiguous viewpoints
>     - Gaze: resolves temporal ambiguity and instrument–verb dependencies
>     - Think: enforces structured, symbolic triplet grounding
>
> **Q2: Strength of Prompts on Performance Improvement**
>
> **A2:** The primary goal of CoS is to elicit the latent capability of LVLMs in the surgical domain **without any additional finetuning**, thereby fully leveraging the increasingly competent LVLMs in uncommon tasks. As shown in Table 2, the LVLM's performance improves progressively under the stepwise Glance–Gaze–Think prompting scheme.
>
> **Q3: Mechanism and ablation of MiST**
>
> **A3:** MiST is designed to address cross-modality inconsistency, while the Machine Encoding branch produces spatial–temporal visual features from the Frame Encoder (SAM + ResNet18) and the Video Encoder (MViT).
>
> As shown in Fig. 2 and Eq. 4~7, MiST uses the **text embeddings** from Glance and Gaze as **queries**, attending to visual features. MiST is not a simple feature concatenation but a cross-modal grounding mechanism that functionally **bridges CoS and Machine Encoding**.
>
> To isolate the contribution of MiST, we added a clear ablation comparing:
>
> - CoS (LVLM reasoning only)
> - Machine Encoding (visual features only)
> - Full system (the MiST-connected CoS and Machine Encoding)
>
> | Method               | $\text{mAP}_i$ | $\text{mAP}_v$ | $\text{mAP}_t$ | $\text{mAP}_{ivt}$ |
> | ------ | :------: | :------: | :------: | :------: |
> | CoS |39.79|36.31|19.79|3.90|
> | Machine Encoding|24.06|19.34|16.36|2.50 |
> | CoS+Machine Encoding |52.54  | 37.56  | 21.55  | 9.39|
>
> **Q4: Model Scale Explanation**
>
> **A4:** Both RDV and RiT use ResNet as their backbone, with model sizes below 200M parameters. HecVL, PeskaVLP, and SurgVLP follow a CLIP-like architecture, and each contains fewer than 1B parameters.
>
> We emphasize that CoCoS is **not intended for real-time intraoperative decision support** but is instead developed for **offline postoperative analysis and workflow evaluation**, where accuracy and interpretability are prioritized. Accordingly, the use of SAM and multi-stage LVLM reasoning reflects a deliberate design choice aimed at maximizing grounding fidelity rather than minimizing inference latency.
>
> The Qwen-VL-Max and QVQ-Max models used in this study are closed-source. To evaluate whether CoS can be effective with relatively smaller LVLMs, we replaced QVQ-Max and Qwen-VL-Max with the open-source Qwen3-VL-8B-Thinking and Qwen3-VL-8B-Instruct models, respectively. The resulting performance comparisons, along with corresponding ablation studies, are summarized below:
>
> | Method| $\text{mAP}_{i}$ | $\text{mAP}_{v}$ | $\text{mAP}_{t}$ | $\text{mAP}_{ivt}$ |
> | ------- | :-------: | :---------: | :--------------: | :----------------: |
> | RDV |   36.90 |10.84|6.91|2.52|
> | RiT (Sharma et al., 2023) |26.75|12.60|12.49|2.02|
> | HecVL (Yuan et al., 2024a)|21.91|17.69|16.43|7.78|
> | PeskaVLP (Yuan et al., 2024b)|      20.10       |      16.23       |      14.14       |        6.72        |
> | SurgVLP (Yuan et al., 2025)|      21.44       |      18.44       |      15.87       |        5.95        |
> | **Ours (Qwen-VL-Max&QVQ-Max)**|    **52.54**     |    **37.56** |    **21.55**     |      **9.39**      |
> | **Ours (Qwen3-VL-8B-Instruct&Thinking)** |    **32.82**     |    **27.67**  |    **22.30**     |      **9.98**      |
>
> | Method | $\text{mAP}_i$ | $\text{mAP}_v$ | $\text{mAP}_t$ | $\text{mAP}_{ivt}$ |
> | ----| :------------: | :------------: | :------------: | :----------------: |
> | Glance|     13.62      |     15.54      |      9.24      |        2.11        |
> | Glance+Gaze|  19.07  |     17.85      |      9.47      |        2.35        |
> | Glance+Gaze+Think  |  19.88  | 19.26  |     10.52      |        2.51        |
> | Glance+Gaze+Think+Machine Encoding |     32.82      |     27.67      |     22.30      |        9.98        |
>
> It indicates that **CoS is capable of effectively eliciting the reasoning abilities of a smaller, open-source LVLM (Qwen3-VL-8B)** for the compositional zero-shot triplet recognition task. Ablation studies further reveal that each reasoning stage provides incremental contributions, with the combination of all stages and Machine Encoding yielding the full performance improvement.
>
> Paper and supplementary have been updated.

---

### Official Review · Reviewer_VZJg · 2025-10-30

**Soundness:** 3
**Presentation:** 3
**Contribution:** 3
**Rating:** 6
**Confidence:** 4

**Summary:**

This paper proposes Cognitive-Awakening Chain-of-Surgery (CoCoS), a framework for compositional zero-shot surgical triplet recognition. It introduces a three-stage Glance–Gaze–Think prompting scheme that mimics human surgical reasoning to enhance large vision-language models’ understanding of surgical videos. A Machine Encoding module provides stable spatial-temporal cues, while the MiST fusion module aligns visual and textual features for robust triplet prediction. A compositional zero-shot split strategy is designed to evaluate generalization to unseen instrument–verb–target combinations. Experiments on CholecT50 demonstrate that CoCoS significantly outperforms existing baselines in accuracy and compositional generalization.

**Strengths:**

1. This work proposes the Cognitive-Awakening Chain-of-Surgery (CoCoS), a three-stage “Glance–Gaze–Think” prompting scheme that mirrors how surgeons observe, analyze, and decide during operations. This staged reasoning introduces an original multimodal adaptation of chain-of-thought, enabling large vision-language models to perform structured and clinically aligned interpretation of surgical videos.

2. This work proposes a new benchmark and data split strategy where each component of a surgical triplet (instrument, verb, target) is seen during training but their combinations are unseen at test time. This setting realistically captures clinical generalization needs and allows systematic evaluation of compositional reasoning in surgical AI.

3. This work proposes the MiST fusion module that effectively aligns spatial (segmentation-based), temporal (video-level), and textual representations, improving model robustness against hallucination. Extensive experiments on the CholecT50 dataset demonstrate clear gains across all triplet components, validating both methodological soundness and practical feasibility.

**Weaknesses:**

1. The method is only tested on CholecT50; it remains unclear whether the framework generalizes to other surgeries or datasets.

2. The multi-stage Glance–Gaze–Think process may be computationally heavy, yet the paper reports no inference time or feasibility for real-time use.

3. Although described as “human-like reasoning,” the work does not measure how well generated reasoning aligns with visual evidence or handles noisy scenes.

Ref:

[1] OphNet: A Large-Scale Video Benchmark for Ophthalmic Surgical Workflow Understanding

[2] OphCLIP: Hierarchical Retrieval-Augmented Learning for Ophthalmic Surgical Video-Language Pretraining

[3] Ophora: A Large-Scale Data-Driven Text-Guided Ophthalmic Surgical Video Generation Model

[4] Towards Dynamic 3D Reconstruction of Hand-Instrument Interaction in Ophthalmic Surgery

**Questions:**

1. Could the authors clarify how the Glance–Gaze–Think prompting is operationalized during inference? For example, is each stage conditioned on the previous stage’s textual output, or are they run independently and then fused by MiST? The paper would benefit from explicitly defining the interaction mechanism.

2. The paper mentions that Machine Encoding stabilizes perception when LVLMs hallucinate, yet it remains unclear how frequent or severe such hallucinations are. Could the authors provide quantitative evidence or failure examples to show how much this module reduces visual hallucination or improves grounding accuracy?

---

> ### Author Response · Authors · 2025-11-25
> **Reply to Reviewer VZJg**
>
> We thank the reviewer for the positive evaluation of the contributions and strengths of our work. We appreciate the constructive comments and address all questions and concerns below.
>
> **Q1: Dataset Generalization**
>
> **A1:** We thank the reviewer for raising the concern about dataset diversity and generalization. We clarify that CholecT50 was the **only** publicly available dataset with large-scale, high-quality triplet annotations at the time of our submission. Therefore, it was the only dataset that allowed constructing a reliable compositional zero-shot split for `<instrument, verb, target>` triplets.
>
> In September this year, the **ProstaTD** [ref1] dataset was released with structured surgical triplet annotations. To directly address the reviewer's point, we conducted an additional compositional zero-shot triplet recognition experiment on ProstaTD. The results are shown below:
>
> | Method| $\text{mAP}_{i}$ | $\text{mAP}_{v}$ | $\text{mAP}_{t}$ | $\text{mAP}_{ivt}$ |
> | ----- | :-----------: | :-------------: | :-------------: | :----------------: |
> | RDV|      34.24       |      10.77       |       7.16       |        2.80        |
> | RiT (Sharma et al., 2023)      |      27.29       |      13.08       |      13.42       |        3.25        |
> | HecVL (Yuan et al., 2024a)               |      16.43       |      14.37       |      14.08       |        6.61        |
> | PeskaVLP (Yuan et al., 2024b)            |      17.62       |      13.62       |      14.03       |        6.88        |
> | SurgVLP (Yuan et al., 2025)              |      15.29       |      15.03       |      15.01       |        5.52        |
> | **Ours (Qwen3-VL-8B-Instruct&Thinking)** |      31.55       |      26.72       |      24.11       |        9.56        |
>
> This additional experiment confirms that:
> - CoCoS is not specific to CholecT50
> - The proposed reasoning framework generalizes across surgical domains (cholecystectomy → prostate surgery)
>
> > Ref:
> > [1] ProstaTD: A Large-scale Multi-source Dataset for Structured Surgical Triplet Detection
>
> **Q2: Inference Time**
>
> **A2:** We thank the reviewer for raising the issue of inference latency. We would like to emphasize that CoCoS is **not intended for real-time intraoperative decision support** but rather for **offline postoperative analysis and workflow evaluation**, where accuracy and interpretability take precedence. Consequently, the use of SAM and multi-stage LVLM reasoning was a deliberate design choice to prioritize grounding quality over latency.
>
> More specifically, **SAM accounts for approximately 85% of the total inference time**, owing to its computationally intensive segmentation backbone. The **LVLM queries (Qwen-VL-Max / QVQ-Max)** issued through the API comprise the majority of the remaining overhead. By contrast, the MiST fusion and Machine Encoding components (ResNet18 + MViT) contribute **less than 5%** to the total runtime.
>
> We concur that real-time deployment is an important avenue for future research. Developing lightweight variants of CoCoS is indeed part of our ongoing work, and we will include a corresponding discussion in the Conclusion section.
>
> **Q3: Reasoning Alignment**
>
> **A3:** The primary purpose of CoS is to activate the inner potential of LVLM in the surgical domain **without any finetuning**, which means fully utilizing the abilities of the increasingly capable LVLM in the uncommon domain. As shown in Table 2, the LVLM's performance improves progressively under the step-by-step Glance–Gaze–Think guidance, indicating that its reasoning becomes increasingly aligned with the visual facts.
>
> **Q4: Glance–Gaze–Think Prompting Details**
>
> **A4:** Multi-turn conversation is applied during the inference. Figure 4 of the supplementary material shows the structure of prompt templates in details. Messages given by the "assistant" role will remind LVLM of former knowledge so that every stage of Glance–Gaze–Think includes all of the previous answers. Finally, in the Think stage of CoS, LVLM will output a formatted judgement of triplet. In MiST, the texts generated during the CoS process serve as queries to retrieve the corresponding visual embeddings.
>
> **Q5: Machine Encoding Influence on Visual Hallucination Reduction**
>
> **A5:** Apologies for the ambiguity in the earlier statement. The comparison between "Glance+Gaze+Think" and "Glance+Gaze+Think+Machine Encoding" in Table 2 quantitatively hints at how hallucination affects performance, which indicates that instrument recognition suffers more heavily from hallucination than the other subtasks.
>
> Paper and supplementary have been updated.
>
> > Ref:
> > [1] OphNet: A Large-Scale Video Benchmark for Ophthalmic Surgical Workflow Understanding
> > [2] OphCLIP: Hierarchical Retrieval-Augmented Learning for Ophthalmic Surgical Video-Language Pretraining
> > [3] Ophora: A Large-Scale Data-Driven Text-Guided Ophthalmic Surgical Video Generation Model
> > [4] Towards Dynamic 3D Reconstruction of Hand-Instrument Interaction in Ophthalmic Surgery

---

### Author Response · Authors · 2025-12-02
**Summary**

## Summary

We thank the Area Chair and the reviewers for their time and constructive feedback. We are encouraged that reviewers recognized the clinical relevance of our proposed task, **Compositional Zero-Shot Surgical Triplet Recognition (CZSTR)**, and the intuitive nature of our **Cognitive-Awakening Chain-of-Surgery (CoCoS)** framework.

During the rebuttal period, we conducted extensive additional experiments to address concerns regarding dataset generalization, model capacity fairness, and component ablation. We provide a summary of our responses and the new evidence below.


### Revisions to Reviewer feedback

**1. Generalization Across Surgical Domains**

**Concern:** Reviewers VZJg and bsRt questioned whether our method generalizes beyond the CholecT50 dataset.

**Response & New Experiment:** We validated our framework on the newly released ProstaTD dataset (prostate surgery).

- **Result:** CoCoS achieved significant improvements over baselines on this completely different surgical procedure.
- **Conclusion:** This confirms that the "Glance-Gaze-Think" reasoning framework is not dataset-specific but captures fundamental surgical semantics applicable across domains.

**2. Model Capacity and Fair Comparison**

**Concern:** Reviewers fhrs and 6Wjd raised concerns that our performance gains might stem primarily from the sheer capacity of large foundation models (Qwen-VL-Max/QVQ-Max), questioning the fairness of comparisons against smaller baselines.

**Response & New Experiment:** To demonstrate that CoCoS is a universal framework capable of activating reasoning potential regardless of model size, we conducted additional experiments using the open-source, smaller-scale **Qwen3-VL-8B**.

- **Result:** The Qwen3-VL-8B model, driven by our CoS framework, achieved significant outperformance, from 2.11 to 9.98 mAP.
- **Conclusion:** These results confirm that our framework effectively elicits the latent capabilities of LVLMs across different scales. The steady performance gains observed through the "Glance-Gaze-Think" stages on the 8B model prove that our contribution lies in the *cognitive reasoning mechanism* that maximizes the utility of the LVLM.

**3. Component Ablations and Technical Validity**

**Concern:** Reviewers fhrs and 6Wjd requested clearer evidence for the contribution of the MIST fusion module and the Chain-of-Surgery (CoS) prompting versus a "Machine-only" approach.

**Response & New Experiment:** We provided "Machine-only"  ablation studies in Table 2.

**4. Inference Latency and Application Scope**

**Concern:** Reviewers VZJg and bsRt noted the computational cost of using SAM and LVLMs.

**Response:** We clarified that CoCoS is designed for offline postoperative analysis (workflow evaluation, education), where accuracy and interpretability prioritize latency. We quantified that SAM consumes ~85% of inference time, while our proposed MIST fusion adds less than 5% overhead. Future work will focus on lightweight distillation for intraoperative use.

### Re-emphasis on the Core Idea of CoS

A central contribution of CoCoS is not architectural complexity but the **ability to activate the latent reasoning potential of LVLMs in the surgical domain without any fine-tuning.** The staged Glance–Gaze–Think process progressively aligns LVLM reasoning with visual evidence, enabling strong compositional generalization that conventional supervised models fail to achieve.


We believe the introduction of the CZSTR task and a proven framework for leveraging LVLMs in surgery offers substantial value to the ICLR community.

Sincerely,

The Authors

---

### Note · Authors · 2026-01-27

I have read and agree with the venue's withdrawal policy on behalf of myself and my co-authors.

---

### Meta-Review · Area_Chair_FcjK · 2025-12-13

**Summary:**

This submission studies compositional zero shot surgical triplet recognition, where each of instrument, verb, and target is seen in training but test requires unseen combinations. Reviewers generally agree the task is important and clinically relevant, and two reviewers (VZJg, bsRt) were initially positive and rated the paper as borderline accept (6), mainly driven by the task framing and the intuitive Glance, Gaze, Think prompting pipeline. In contrast, two senior and more skeptical reviewers (6Wjd, fhrs) rated it as reject (2), focusing on limited ML novelty, large capacity mismatch due to heavy use of off the shelf foundation models, and missing ablations and training details.

The rebuttal is substantial and clearly improves the paper. The authors added cross domain validation on ProstaTD, clarified what is trained and how SAM masks are selected, added a machine only baseline, and provided capacity related evidence using an open 8B LVLM. They also clarified the intended offline use case and gave a rough runtime breakdown. Reviewer bsRt stayed positive in discussion, while reviewer 6Wjd explicitly maintained the reject score after reading the rebuttal, stating that the work remains an engineering pipeline leveraging strong LVLMs without a clear new learning principle.

After considering the updated evidence and the discussion, I appreciate the effort and I agree the task and empirical gains are interesting. Still, my main issue aligns with the key concerns: the core technical contribution looks like a careful orchestration of existing components (SAM, encoders, LVLM prompting, and a fusion module), and it is not yet clear what transferable ML insight the community should take away beyond “prompt and combine stronger models carefully.” Given the ICLR bar and the split reviewer signal, I recommend rejection.

**Reviewer Concerns:**

Concerns largely addressed by the rebuttal:

1. Added ProstaTD results and showed the framework transfers reasonably.

2. Added machine only baseline and clearer component ablations, including the role of MiST and Machine Encoding.

3. Clarified what modules are trained and provided more details on SAM mask selection and objectives.

4. Added experiments with an open Qwen3 VL 8B variant to show the staged prompting can help beyond closed models.

Concerns that remain outstanding:

1. Even with stronger experiments, the method still reads as an engineered pipeline built around strong foundation models, with limited new algorithmic or learning insight. This was reiterated by reviewer 6Wjd after the rebuttal.

2. The added 8B experiment helps, but it does not fully resolve the concern that performance is dominated by access to powerful LVLMs and heavy components, and it remains unclear what a simpler LVLM only baseline with comparable prompting would achieve.

3. Some details improved, but the approach remains expensive and depends on external APIs, and the promised code release is post acceptance.

**Reviewer Scores:**

Reviewer VZJg: likely stays at 6. Concerns on generalization and latency were addressed, and this reviewer was already positive.

Reviewer bsRt: stays at 6. The reviewer explicitly kept the positive recommendation in discussion.

Reviewer fhrs: likely remains negative, since their main concern is that CoS is not a meaningful community contribution and MiST was initially under motivated. The rebuttal helps but may not change their overall stance.

Reviewer 6Wjd: stays at 2. The reviewer explicitly reiterated that the core novelty remains limited and retained the original score.

Overall, while the rebuttal improves clarity and strengthens the empirical story, the novelty concerns remain decisive for me, so my final decision is reject.

---

### Decision · Program_Chairs · 2026-01-26

Reject